# Factors affecting contraceptive choice in women over 40: a qualitative study

Jo Burgin  ,[1] Julia V Bailey  [2]

[1]Centre for Academic Primary Care, Bristol Medical School, Bristol, UK
[2]eHealth Unit, Department of Primary Care and Population Health, University College London, London, UK

**Correspondence to**
Dr Jo Burgin; jo.burgin@nhs.net

## ABSTRACT

**Objective** To explore the views of women over 40 years in choosing and using contraception, and to inform how contraceptive counselling for this age group could be improved.

**Design, setting and participants** Fourteen women aged 40–52 years were recruited through social media platforms to take part in online, semistructured, in-depth interviews. Transcripts were analysed using a qualitative thematic approach.

**Results** (1) Participants were anxious about unplanned pregnancy, and still highly motivated to avoid this. (2) Changes of contraceptive method over the lifecourse were occasionally precipitated by emergent health conditions, but healthcare providers often recommended a change in method on the basis of age alone. (3) Participants were experiencing perimenopausal symptoms but were largely unaware of how hormonal contraception could be used to treat these symptoms. (4) Prior negative experiences with contraceptive methods, coercive experiences with healthcare providers, and traumatic life events all contributed to a narrowing of contraceptive preference in later life.

**Conclusion** Women over 40 years may be highly motivated to avoid pregnancy. This age group may have complex contraceptive histories with emerging perimenopausal symptoms. Women over 40 years may have accumulated adverse experiences which impact their contraceptive choices. These factors need to be explored by clinicians, to facilitate shared decision-making.

## INTRODUCTION

Motivators and barriers to contraception use change over the course of a woman's life.[1] Changes in motivation to avoid future pregnancies, medical history and perimenopausal symptoms mean that women over 40 years may need to review their contraception method of choice. Historically, contraceptive counselling has been poorly defined, and has concentrated on the clinician imparting information to provide the patient with a safe, acceptable and reliable method. However, more recently, there have been attempts to define contraceptive counselling as a dynamic 'exchange of information' between the patient and the provider, which puts patients' needs and preferences at the heart of the consultation.[2] The majority of women over

40 years are provided with contraception in primary care.[3] Lack of flexibility in appointment time, and the increasing popularity of templates in general practice may lead to a 'one-size-fits-all' approach, which neglects the complex needs of older women.[4–6]

### Preventing pregnancy

Most women aged 45–49 years are sexually active, and 72% report using a contraceptive method.[7] However, women aged 40 years and over have one of the highest rates of abortion compared with live births. In 2019, 31% of pregnancies in this age group ended in therapeutic abortion, which is higher than women under 20 years.[8] This suggests there is continuing unmet need, or that women over 40 years may misjudge the likelihood of pregnancy.[9]

### Increased health risks and symptom control

Background risks of cardiovascular disease, breast cancer and other gynaecological cancers increase with age.[10] This may affect

choice and clinical eligibility for certain contraceptive methods, particularly combined hormonal methods. As women enter the perimenopause, they can experience changes to their menstrual cycles or perimenopausal symptoms, which may influence their method choice.[11]

### Previous contraceptive experience

On average, women will try approximately three different contraceptive methods over the course of their reproductive lifetime.[12] Women who have had negative experiences with contraception are more likely to report unintended pregnancy in later reproductive life.[13]

Research in women over 40 years has been heavily focused on which methods are clinically safe and effective. Globally, quantitative data on contraception are either not collected or are routinely incomplete for women over 45 years and may neglect to record intrauterine device (IUD) use or previous hysterectomy.[14] Qualitative research has focused on young women's experiences of contraception but neglected the specific priorities and concerns of older women.[15] This paper aims to explore the views of women over 40 years in choosing and using contraception, to inform how contraceptive counselling for this age group could be improved.

## METHODS
### Participants, setting, recruitment

Women aged 40–55 years were recruited from adverts on social media platforms (Twitter and Facebook) between June and November 2021. Twitter adverts were placed on the accounts of the research team and their academic departments; these were then disseminated by other academics and organisations with interests in contraception (eg, Faculty for Reproductive and Sexual Health, Decolonising Contraception), endometriosis (eg, Endobonds) and menopause (eg, Menopause Café, Black Women in Menopause). Adverts were also placed on Facebook pages for Perimenopause and Menopause interest groups. Finally, a study page was set up on the research participant website 'Call for Participants', which recruited participants between June and December 2021. Four participants were forwarded information from social media adverts by colleagues or friends.

Twenty-three participants made contact with the primary researcher and 14 proceeded to interview following receipt of the study information leaflet and the consent form. Recruitment took place during the COVID-19 pandemic when access to healthcare was limited by intermittent lockdowns and reduced National Health Service capacity, particularly for non-urgent care.

### Data collection and analysis

Online, semistructured, in-depth interviews were conducted between August and December 2021. One-to-one interviews lasted 45–60 min and were carried out by JB, who is also a primary care doctor. A discussion guide was used to cover broad areas of interest (see online

supplemental appendix 1). Participants had the choice to have their cameras on or off for the duration of the interview: 12 participants chose to have cameras on, while 2 chose to have them turned off. The researcher had their camera on for all interviews. Interviews were recorded on Microsoft Teams, transcribed using computer software and subsequently checked against original recordings for accuracy. Transcripts were uploaded to NVivo V.12 software for coding and thematic analysis.[16]

A combination of inductive and deductive thematic analysis was used.[17] All interviews were read a minimum of three times; the first reading identified broad themes, which were largely influenced by the topic guide. The second reading allowed for a more inductive approach, where data were coded line by line and broad, or significant themes were reworked. Finally, a third reading cross-referenced themes with the data to ensure correlation and depth of meaning. Participant recruitment continued until no new themes were identified.

### Patient and public involvement

Participants' views shaped new content on contraception for women over 40 years on the Contraception Choices website. All participants will receive a copy of any published work.

## RESULTS
### Participant characteristics

Fourteen cis-gendered women aged 40–52 years (average age 45 years) were recruited from across England, with the majority located in Southern England. All women were employed, or lived with a partner who was employed, were digitally literate and did not require the assistance of a translator to complete their interviews. Table 1 shows selected sexual and reproductive health characteristics of the participants.

### Themes

Participants discussed their perceived pregnancy risk, how their healthcare needs had changed over their life course and disclosed negative experiences which affected their approach to contraception.

### Fear of pregnancy

Fear of pregnancy remained a key motivator in continued contraception use. Participants perceived an unintended pregnancy as a major negative life event.

Despite declining fertility in women over 40 years, almost all the participants felt pregnancy was a real possibility and remained cautious in their sexual relationships. Some mentioned women in their late 40s and 50s becoming pregnant unexpectedly. These narratives seemed to emphasise the real possibility of pregnancy, which they identified with.

> Yeah, and I have heard of women that have naturally conceived twins in their 50s. OK, maybe they've been with a, probably, 20–30 year old at the time with

**Table 1** Sexual and reproductive characteristics of participants

| Age | |
|---|---|
| Range | 40–52 years |
| Mean | 45 years |
| **Relationship status** | |
| Single | 2 |
| Different gender | 11 |
| Same gender | 1 |
| **Children (biological)** | |
| Yes | 10 |
| No | 4 |
| **Experiencing perimenopausal symptoms** | 8 |
| **Current use of HRT** | 6 |
| **Current contraceptive method** | |
| Intrauterine system | 4 |
| Progestogen-only pill | 3 |
| Implant | 2 |
| Copper intrauterine device | 1 |
| Condoms | 1 |
| Vasectomy | 1 |
| None | 2 |
| **Previous number of contraceptive methods used** | |
| 2–3 | 8 |
| 4–5 | 5 |
| 6+ | 1 |
| **Location (England)** | |
| South East & London | 7 |
| South West | 6 |
| North West | 1 |

HRT, hormone replacement therapy.

my one example, but it could happen, couldn't it? (Participant 2, age 50)

The majority of participants had completed their families and mentioned the stress an additional child would place on, not just themselves, but the rest of the family. Some participants had both children and parents as dependents, or were at a demanding juncture in their career, and so did not want to extend their responsibilities with another child. Finances had also played into decisions to limit family size. Other participants did not want to become parents and knew they would have to terminate a pregnancy if their contraceptive method failed.

No, I think contraception is important. Because for example, if I had another child now, at this stage in life, number one, I certainly couldn't cope. I mean, I know I need to sleep. You know I have an elderly parent that I've helped with, and when you're up all

night, that's not something I can do in my late 40s. (Participant 11, age 48)

Some participants mentioned the negative physical effects of pregnancy and traumatic birth experiences as a motivator in using contraception to prevent pregnancy, and also as increasing their level of anxiety over the possibility of contraceptive failure.

…and the way that my labour went, and my induction went and everything, I don't think I could put myself through that again. I mean there were just so many things that happened and then went wrong. (Participant 10, age 43)

Worry around unplanned pregnancy created anxiety for women and their partners. Some women described their partners checking they had taken their pills, or distrust of methods they were not familiar with, like the intrauterine system (IUS). For participants not in long-standing relationships, this anxiety impacted their enjoyment of new relationships.

This should be fun, and actually it's just made it more scary. Petrified I was going to get pregnant again which would be horrendous…That would be a moral dilemma, from which I probably wouldn't recover because I couldn't do another child. (Participant 8, age 52, on starting a new relationship in her 50s)

Almost every participant in a long-term relationship had discussed the possibility of vasectomy with their partner, but the majority said that their partners were firmly opposed. This led to a feeling that the responsibility for contraception was one sided, and that this was an additional burden that they had shouldered over the lifetime of their relationships.

Didn't even draw breath. Just went, "No"! Fair enough. I can't make him. You know that would have been my preferred option…I felt like I've just done three rounds of IVF [in vitro fertilisation] and two children and I don't want any more chemicals in my system. And he was like, No. (Participant 14, age 41)

Only two participants said that their partner took responsibility for contraception and only one participant was currently relying on vasectomy as a form of contraception.

### Change of contraceptive methods over the life course
All but one participant had used the combined contraceptive pill (COCP) in their earlier contraceptive years. Increasing age, health conditions and side effects had influenced method changes, with the majority of participants reporting current use of progestogen-only methods.

### Age
Many participants had started the COCP as teenagers as their first form of contraception. While some stopped taking it due to side effects, others continued to take it for decades without issue. However, many participants felt

that being on hormonal contraception for long periods of time was harmful and that they either tried to take regular breaks or, as they got older, they became more concerned about associated health risks. For some, this was highlighted by doctors who advised method changes in their late 30s and early 40s. For others, hearing about the negative experiences of friends of colleagues influenced their decision to switch methods.

> Well, you know with the contraceptive pill, you always know that there are some risks, like you could have a stroke, or you could have a thrombosis. Or, you know, there are risks of cancer. But when you were starting out, should we say in your 20s, these risks seems small. But when you become older and you've seen people that might have gone through this, so you know they might now be disabled because they've had the thrombosis…and then the doctor says you are of an age where you shouldn't continue with contraceptive pills, you listen. (Participant 11, age 48)

Many women accepted this change in contraception as part of a natural progression in their contraceptive use, and a switch to a progestogen method or a coil was relatively seamless. However, for some women, advice to change method seemed arbitrary and very little explanation had been provided prior to stopping the COCP.

> I think my doctor had said previously. He's like at some point, you know we'll talk about maybe switching to progesterone only pill because you, um, "you're at that age now", or some kind of like throwaway statement that makes you go "What? What does that mean?" (Participant 4, age 43)

For those who had been satisfied with the COCP, stopping it resulted in the unmasking of perimenopausal symptoms.

> I thought oh it can't be good for me to have all these hormones in me, maybe I should try and reduce that a little bit. Turns out my body really liked the hormones! (Participant 4, age 43)

While all participants were offered progestogen-only methods or IUDs in place of the COCP, some experienced significant progestogen-related side effects such as irregular bleeding, mood changes, breast tenderness and bloating. Participants also reported difficulty accessing coil insertion if this was not offered in their general practitioner surgery. This meant navigating coil providers at community hospitals and sexual health clinics, with some participants reporting they had been turned away due to their age.

### Switching methods

Participants had tried between two and six contraceptive methods over their lifetime. Those who had used more than three methods reported hormonal side effects as a reason for switching methods, and this made finding suitable contraception in later life problematic. Mood changes were the most commonly reported concern when trying different hormonal methods.

> I'd previously been on the injection…hormone…thing, for contraception, some years previously and it sent me a little bit like, you know, insane, I can't think of a better word. So I questioned them quite heavily about that side of things because I wasn't keen to go through that again. (Participant 5, age 41)

Painful coil insertions (both personal experiences and narratives from friends or family) were frequently cited as reasons for discounting intrauterine contraception in later life.

> I'm a bit scared to have it put in because everybody that I've spoken to who has one, or has had one, says it really hurts. So I'm just like, I hurt enough round there. (Participant 7, age 50)

### Lack of tailored support from healthcare providers

Some participants were confused at the variety of methods that had become available since they started using contraception and did not feel supported in making new decisions about contraception methods. This was compounded by a feeling that they had individual needs that were not being taken into consideration. In fact, at a time where their medical history and personal preferences were most complex, it was felt they were being treated as a homogeneous group based solely on age.

> Whereas in kind of (my) 20s and 30s I did feel very generic… I'm looking for contraception to have a different effect than what I would have done before. (Participant 3, age 42)

### Health conditions

Two participants mentioned health conditions (migraines and high blood pressure) that precipitated a change from the COCP. Both reported problems with alternative methods of contraception and relied for some period of time on withdrawal methods, which, for one participant, resulted in an unintended pregnancy.

However, other participants had health conditions which had led to a change of contraception specifically for its health benefits. Endometriosis and menorrhagia were both drivers for considering the IUS. Another participant started a progestogen method to tackle premenstrual syndrome.

### Perimenopause

Confusion about the interplay between the menopause and hormonal contraceptive methods was common. Participants did not understand how they would identify the menopausal transition while taking contraceptive methods that already affected their periods. This caused frustration as it could mean they were taking contraception even when it may no longer be needed to prevent pregnancy.

And I would love not to be on the pill. I'm assuming I'm gonna hit menopause soon. More than happy to. I'm embracing it. And it really is like, bring it on. Let me get on with the rest of my life. But how do you know, if you're taking the pill? How on earth do you know? (Participant 2, age 50)

Many participants spoke about perimenopausal symptoms and the difficulties they had experienced diagnosing and treating their symptoms. While some participants had been offered the IUS to control heavy periods, or as an alternative to the COCP, few were aware that the IUS could be used as a component of hormone replacement therapy (HRT). Only one participant knew that combined hormonal contraception could alleviate perimenopausal symptoms. Overall, there was a sense of frustration that their hormonal health had not been treated more holistically and that perimenopausal symptoms were misdiagnosed or ignored.

I don't think anyone has ever really spoken to me about how you might want to change contraception as you do approach menopause and why. (Participant 12, age 48)

### Previous experiences of coercion and trauma

Experiences of contraceptive coercion in a medical setting had severe and long-lasting effects. These incidents left participants feeling a loss of control over their own bodies and sense that their wishes had been violated. This undermined trust in both medical professionals and medical treatments. Traumatic experiences influenced the type of contraception that participants felt comfortable with, namely a method that was under complete user control such as condoms and pills. It also made participants hesitant about any methods that involve intimate procedures where they may experience a similar loss of autonomy and flashbacks. Complex and traumatic birthing experiences were also reasons for declining intrauterine contraception or for requiring sedation in order to have one fitted.

Yeah, I hated that I felt really violated by almost, almost instantly. I felt like I'd been massively pressurized into it, and it was. I couldn't get it out again…I felt yeah really out of control…. And I've always been, since I had that injection, terrified of them. Anything hormonal. (Participant 13, age 40, discussing contraceptive injection as a university student)

Yeah, I mean even going for a smear is a big thing. I cried through most of them just because you're in that position. Somebody is going there. It's gonna hurt. I know they say it doesn't, but it, it does. I probably cry about it now, which is ridiculous, isn't it? (Participant 8, age 52, following failed induction of labour)

Conversely, a few participants reported their repeated exposure to medical procedures as enabling them to manage more intimate procedures, like the insertion of an IUD, with ease. The stark difference between these experiences seemed to centre around how much autonomy the participant felt they had during these procedures.

I had no problems, but I've done three rounds of IVF so I had so…sounds horrendous!…so many things put inside me, that actually this wasn't an issue. (Participant 14, age 41)

Other traumatic incidents outside the medical setting, including sexual assault and near miss attacks, were also reported and influenced choice of contraception. Again, this reinforced the need for control over their body and reproductive choices.

Yeah, I've had another really traumatic experience when I was 16 when somebody almost raped me and I think maybe I felt like it was really important to be totally in control. (Participant 13, age 40)

## DISCUSSION

Women over 40 years have distinct motivators and barriers to choosing and using contraception. Previous studies have attributed the relatively low use of reliable contraception in women over 40 years to women's perceptions of themselves as low risk due to declining fertility.[9 13] However, this study found that while women recognised that fertility levels decline in their 40s, they thought of pregnancy as a real possibility, and something that would have a large and detrimental impact on their lives. It created anxiety both for them as individuals and also in their relationships.

Many participants reported stopping the COCP on the basis of age alone. This seemed to be provider led, but is not supported by clinical guidelines. In fact, combined hormonal contraception is increasingly recognised as a useful tool in women over 40 years as it provides both contraception and stable levels of oestrogen, thereby treating perimenopausal symptoms.[18] Participants valued a tailored approach to contraceptive counselling and disliked being advised solely on the basis of age, which echoes the advice in clinical guidelines.[4 19–21] Despite this, women still feel they are not treated either holistically or as individuals.

Many participants felt frustrated and disappointed that their healthcare provider had not helped them to identify perimenopausal symptoms earlier. Perimenopausal symptoms are not routinely discussed in contraception reviews, meaning that these are missed, especially if women's periods are already altered by hormonal contraception.[22] Participants felt they would benefit from earlier information on the perimenopause and HRT in order to inform their contraceptive decisions.

Women over 40 years are usually experienced in using contraception, but sometimes it is these experiences which create further obstacles to satisfactory contraception choice and use. Side effects from previous contraception may make women more apprehensive to try new

methods,[23–25] which is pertinent for women over 40 years as they have had more exposure to adverse experiences over their reproductive lifetime. Interactions with healthcare providers in all areas of sexual and reproductive health coloured the contraception methods that women felt comfortable with. Experiencing coercion in a healthcare setting led to particularly strong negative feelings and narrowed contraceptive choices. These experiences undermined trust in methods that require provider insertion, which are typically the most long acting and reliable. Traumatic experiences outside a healthcare setting, such as sexual assault, also influenced women's method choice.

Online interviews allowed recruitment from areas of England and participants were able to participate across multiple healthcare localities and providers. However, online recruitment excluded women without access to the internet, and recruitment through social media platforms narrowed the range of potential participants. This may have oversampled women with particularly difficult contraceptive needs or experiences, and those with higher anxiety about risk of pregnancy. Recruitment through specific social media platforms and interest groups may have also led to a relatively homogeneous participant group that limited variety of themes discussed. Recruitment strategies to identify women with no digital access or who had additional needs such as translation may have provided additional insights. The interviews and analysis were conducted by a researcher with a clinical background, which may have been beneficial in interpreting their contraceptive history, but could have influenced the type of information women were comfortable sharing. Some participants may have been hesitant to share their negative experiences with healthcare providers or 'non-medical' approaches to contraception, for example, use of withdrawal methods or fertility tracking. The interviewer was a woman of comparable age with the participants, which may have facilitated open discussion regarding personal experiences.

Clinicians and policymakers should recognise that contraception is still a priority for many women over 40 years. Contraceptive options should be assessed based on individual needs and risk factors, rather than by age alone. It is particularly important to engage in meaningful discussions about the risks and benefits of continuing with a combined hormonal method so that contraceptive options are not narrowed arbitrarily.[26] For women over 40 years, a reframing of contraception within a 'hormonal health' consultation could allow more space for women to share their particular concerns and needs. This would provide an opportunity to discuss the perimenopause and which methods could provide both contraception and relief of symptoms. Allotting appropriate time for contraceptive counselling would allow clinicians to ask about past experiences with contraceptives and previous traumatic experiences, both of which may influence the patient's contraceptive choice. These barriers are not traditionally addressed during contraceptive counselling.[27] While they are clearly important issues to address in every age group,

it could be seen as particularly pertinent for women over 40 years who are more likely to have been through traumatic births and invasive gynaecological investigations. Enquiring about these types of experiences when initiating or switching contraception would be a positive step in making the consultation more tailored and holistic.

Areas of future research could focus on how we meet the specific needs of women over 40 years in practice, including how best to discuss risks and benefits of COCP with women over 40 years; how we can integrate perimenopause and contraceptive counselling effectively; and whether the principles of trauma-informed care can contribute to contraceptive counselling to widen the contraceptive choices for women over 40 years.

## CONCLUSION

Women over 40 years may be highly motivated to avoid pregnancy. This age group may have complex contraceptive histories with emerging perimenopausal symptoms. Women over 40 years may have accumulated adverse experiences, which impact their contraceptive choices. These factors need to be holistically explored by clinicians, to facilitate shared decision-making.

**Contributors** JVB and JB devised the study. JB designed the topic guide with input from JVB. JB carried out all interviews, transcription and analysis with supervision from JVB. Emergent results were discussed with other researchers. JB wrote the article with input from JVB. Both authors approved the final version and are accountable for all aspects of the work. JB is the overall content guarantor.

**Funding** JB is a National Institute for Health Research academic fellow.

**Competing interests** None declared.

**Patient and public involvement** Patients and/or the public were involved in the design, or conduct, or reporting, or dissemination plans of this research. Refer to the Methods section for further details.

**Patient consent for publication** Not required.

**Ethics approval** This study involves human participants and was approved by the University College London (UCL) Ethics Committee (reference number 19875/001). Participants gave informed consent to participate in the study before taking part.

**Provenance and peer review** Not commissioned; externally peer reviewed.

**Data availability statement** Data are available upon reasonable request. Fully anonymised interview transcripts are available from Dr Jo Burgin (jo.burgin@nhs.net). Available for research purposes only.

**ORCID iDs**
Jo Burgin http://orcid.org/0000-0003-0970-244X
Julia V Bailey http://orcid.org/0000-0002-5001-0122

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
