## [Reviewer comments · BMJ Open]

ARTICLE DETAILS

TITLE (PROVISIONAL)	Factors affecting contraceptive choice in women over 40: A qualitative study
AUTHORS	Burgin, Jo; Bailey, Julia

VERSION 1 – REVIEW

REVIEWER	Rezel-Potts, Emma KCL
REVIEW RETURNED	28-Jun-2022

GENERAL COMMENTS	Overall: A qualitative analysis exploring views of women over 40 on choosing and using contraception. Thank you for this well-executed, concise and insightful paper into a much neglected, but hugely pertinent area of reproductive health. I have only minor comments and suggestions. Abstract: - I would slightly reword the conclusion within the abstract to avoid making statements which imply generalisability of your findings. Background: - The background could do with some further context on the timing and setting of the research, i.e. that this was a period of healthcare services being both underfunded and in high demand with difficulties getting appointments and services further exacerbated by pandemic lockdowns. Methods: - The methods are a little sparse. Were these one-to-one interviews conducted entirely online? Did the researcher and participant have cameras on or off? Was a topic guide used and if so can this be included in the publication? Was analysis entirely inductive? - Are you familiar with some of the controversy around defining “saturation”? See Sebele-Mpofu 2020 and Saunders et al. 2018. Fourteen interviews seems quite low to reach saturation of themes. Do you think a more diverse sample or a greater range of recruitment techniques may have altered your saturation point? - In your table, please include the unit years for the age rows. I would also be interested in seeing a further breakdown for previous number of contraceptive methods used, e.g. 0; 1-3, 4-6, 6-8. Could also consider including percentages. Do you have any other sociodemographic information, e.g. geographic location, deprivation, ethnicity?
---

	- I like that there is some self-reflexive commentary on the author's role as a clinician potentially influencing the interviews. However, was this the interviewer's most salient trait as perceived by participants? What about age and gender? Results:  - The subsections "change of contraceptive methods over the lifecourse" and "previous experiences of contraception, coercion and trauma" contain a lot of overlap so could be merged. - Avoid language of probability more appropriate for quantitative research. Please reword the sentence beginning, "Those that had used more than 3 methods were more likely to report hormonal side effects..." Discussion:  - The discussion is good, but the structure is difficult to follow. I advise following convention (and BMJ Open guidance) and restructure into: statement of principal findings; strengths and weaknesses of the study; strengths and weaknesses in relation to other studies, discussing important differences in results; the meaning of the study; possible explanations and implications for clinicians and policymakers; and unanswered questions and future research.
--	--

REVIEWER	Moulton , Jessica E. The University of Melbourne
REVIEW RETURNED	15-Jul-2022

GENERAL COMMENTS	Thank you for this very interesting study and important work on contraceptive choices in women. I thoroughly enjoyed reading the paper and feel that this study could be a valuable addition to the literature. Please find attached some suggested revisions: Strengths and limitations  - Perhaps could list another strength here? Though the authors have done a great job with noting some of the key limitations with online/social media recruitment Introduction  - Line 7, as recommendations to improve contraceptive counselling is one of the key outcomes, it might be worthwhile providing a definition here i.e. what is current best practice for contraceptive counselling, and who provides it – is it doctors and/or nurses, or any health practitioner providing information to people on contraceptive options, etc. Especially as your results so well describe that efficacy is often not the most important factor for people when choosing contraception, and so providing a definition in the introduction would highlight this disparity between current practice and patient preference. - "Research in women over 40 has been heavily focused on what methods are clinically safe and effective" – very important point. - I think a couple of summary statements are needed towards the end of the introduction to strengthen this section. The first paragraph alludes to the fact that current practice may not be appropriate for women over 40, and that women over 40 have complex needs, however if there is more evidence of contraceptive
---

	counselling being insufficient this would be good to add. The definition for contraceptive counselling as above, would also provide further clarity Methods  - This section would benefit from some more detail. For example, under participants, setting and recruitment: what type of social media platforms were used?, what kind of Facebook groups were posted in (women’s health groups, targeted advertising, etc)? - For data collection and analysis, what kinds of questions were asked? A summary of topics or question types would be great here - Also, how did thematic analysis occur, was it independently or with 2 authors? Inductive or deductive coding? - Please also provide a reference for thematic analysis - A reflexivity statement would benefit the methodology section also. The author highlights their role as a clinician researcher, and how this may have influenced the results in the discussion section, however it would be beneficial to elaborate on this within the methods section to understand the authors lens/views on contraceptive counselling, how the results may have been skewed (what information may patients not have shared/etc) and what the authors did to try to minimise this risk. Discussion  - Really interesting results and discussion section. Some great recommendations for future practice, however it would be great to see these summarised at the end of the section – perhaps in the conclusion? I would also be interested to see recommendations for policy/guidelines and future research if any were apparent. Thank you again for the opportunity to review this very interesting research. I look forward to reading a revised manuscript.
--	--

VERSION 1 – AUTHOR RESPONSE

Reviewer 1	
I would slightly reword the conclusion within the abstract to avoid making statements which imply generalisability of your findings.	Completed.
The background could do with some further context on the timing and setting of the research, i.e. that this was a period of healthcare services being both underfunded and in high demand with difficulties getting appointments and services further exacerbated by pandemic lockdowns.	Completed. This has been placed in the methods section as it seemed more relevant to the situation in which the research was conducted than the background of the issue itself.
The methods are a little sparse. Were these one-to-one interviews conducted entirely online? Did the researcher and participant have cameras on or off? Was a topic guide used and	Additional detail added. Topic guide provided as supplementary file.

if so can this be included in the publication? Was analysis entirely inductive?	
Are you familiar with some of the controversy around defining “saturation”? See Sebele-Mpofu 2020 and Saunders et al. 2018. Fourteen interviews seems quite low to reach saturation of themes. Do you think a more diverse sample or a greater range of recruitment techniques may have altered your saturation point?	Agree with this – have included more text on this in methods section.
In your table, please include the unit years for the age rows. I would also be interested in seeing a further breakdown for previous number of contraceptive methods used, e.g. 0; 1-3, 4-6, 6-8. Could also consider including percentages. Do you have any other sociodemographic information, e.g. geographic location, deprivation, ethnicity?	We have not included percentages as these could be misleading with such small denominators. Location data added. We do not have data on deprivation or ethnicity.
I like that there is some self-reflexive commentary on the author’s role as a clinician potentially influencing the interviews. However, was this the interviewer’s most salient trait as perceived by participants? What about age and gender?	Added detail on this in benefits and limitations of study in discussion.
The subsections “change of contraceptive methods over the lifecourse” and “previous experiences of contraception, coercion and trauma” contain a lot of overlap so could be merged.	Moved previous contraceptive experience to change of methods over the life course. We feel that coercion and trauma stands as a distinct experience that warrants separate section.
Avoid language of probability more appropriate for quantitative research. Please reword the sentence beginning, “Those that had used more than 3 methods were more likely to report hormonal side effects...”	Reworded.
The discussion is good, but the structure is difficult to follow. I advise following convention (and BMJ Open guidance) and restructure into: statement of principal findings; strengths and weaknesses of the study; strengths and weaknesses in relation to other studies, discussing important differences in results; the meaning of the study: possible explanations and implications for clinicians and policymakers; and unanswered questions and future research.	Restructured to improve readability and fit more with convention:  • Discussion is structured into paragraphs which start with a principal finding, followed by their relation to other studies (where applicable) and their meaning within the study. • The benefits and limitations are then discussed, followed by implications for clinicians and policymakers and unanswered questions and future research.
Reviewer 2	

Perhaps could list another strength here? Though the authors have done a great job with noting some of the key limitations with online/social media recruitment	Completed.
Line 7, as recommendations to improve contraceptive counselling is one of the key outcomes, it might be worthwhile providing a definition here i.e. what is current best practice for contraceptive counselling, and who provides it – is it doctors and/or nurses, or any health practitioner providing information to people on contraceptive options, etc. Especially as your results so well describe that efficacy is often not the most important factor for people when choosing contraception, and so providing a definition in the introduction would highlight this disparity between current practice and patient preference.	Completed – with references to recent WHO policymaker definition of contraceptive counselling. While there is evidence that women over 40 mainly go to their GP for contraception from NATSAL surveys, there is no data on which clinicians within primary care are providing contraceptive counselling.
I think a couple of summary statements are needed towards the end of the introduction to strengthen this section. The first paragraph alludes to the fact that current practice may not be appropriate for women over 40, and that women over 40 have complex needs, however if there is more evidence of contraceptive counselling being insufficient this would be good to add. The definition for contraceptive counselling as above, would also provide further clarity	Completed. There is a lack of data for contraceptive use and counselling in this age group.
This section would benefit from some more detail. For example, under participants, setting and recruitment: what type of social media platforms were used?, what kind of Facebook groups were posted in (women's health groups, targeted advertising, etc)?	Completed.
For data collection and analysis, what kinds of questions were asked? A summary of topics or question types would be great here	Topic guide provided as supplementary file.
Also, how did thematic analysis occur, was it independently or with 2 authors? Inductive or deductive coding?	Additional detail added to methods section.
Please also provide a reference for thematic analysis	Completed.
A reflexivity statement would benefit the methodology section also. The author highlights their role as a clinician researcher, and how this may have influenced the results in the	More detail added on reflexivity in discussion section (strengths and limitations). As this section is also included at the beginning of the paper we felt this would be too much repetition

discussion section, however it would be beneficial to elaborate on this within the methods section to understand the authors lens/views on contraceptive counselling, how the results may have been skewed (what information may patients not have shared/etc) and what the authors did to try to minimise this risk.	within the article to also include in the methods section.
Really interesting results and discussion section. Some great recommendations for future practice, however it would be great to see these summarised at the end of the section – perhaps in the conclusion? I would also be interested to see recommendations for policy/guidelines and future research if any were apparent.	Restructured to provide recommendations for clinicians and future research in own paragraphs in discussion section.

VERSION 2 – REVIEW

REVIEWER	Rezel-Potts, Emma KCL
REVIEW RETURNED	08-Sep-2022
GENERAL COMMENTS	All comments have been adequately addressed.